# IMAGE SEGMENTATION BY ITERATIVE INFERENCE FROM CONDITIONAL SCORE ESTIMATION

## ABSTRACT

Inspired by the combination of feedforward and iterative computations in the visual cortex, and taking advantage of the ability of denoising autoencoders to estimate the score of a joint distribution, we propose a novel approach to iterative inference for capturing and exploiting the complex joint distribution of output variables conditioned on some input variables. This approach is applied to image pixel-wise segmentation, with the estimated conditional score used to perform gradient ascent towards a mode of the estimated conditional distribution. This extends previous work on score estimation by denoising autoencoders to the case of a conditional distribution, with a novel use of a corrupted feedforward predictor replacing Gaussian corruption. An advantage of this approach over more classical ways to perform iterative inference for structured outputs, like conditional random fields (CRFs), is that it is not any more necessary to define an explicit energy function linking the output variables. To keep computations tractable, such energy function parametrizations are typically fairly constrained, involving only a few neighbors of each of the output variables in each clique. We experimentally find that the proposed iterative inference from conditional score estimation by conditional denoising autoencoders performs better than comparable models based on CRFs or those not using any explicit modeling of the conditional joint distribution of outputs.

## 1 INTRODUCTION

Based on response timing and propagation delays in the brain, a plausible hypothesis is that the visual cortex can perform fast feedforward (Thorpe et al., 1996) inference when an answer is needed quickly and the image interpretation is easy enough (requiring as little as 200ms of cortical propagation for some object recognition tasks, i.e., just enough time for a single feedforward pass) but needs more time and iterative inference in the case of more complex inputs (Vanmarcke et al., 2016). Recent deep learning research and the success of residual networks (He et al., 2016; Greff et al., 2016) point towards a similar scenario (Liao & Poggio, 2016): early computation which is feedforward, a series of non-linear transformations which map low-level features to high-level ones, while later computation is iterative (using lateral and feedback connections in the brain) in order to capture complex dependencies between different elements of the interpretation. Indeed, whereas a purely feedforward network could model a unimodal posterior distribution (e.g., the expected target with some uncertainty around it), the joint conditional distribution of output variables given inputs could be more complex and multimodal. Iterative inference could then be used to either sample from this joint distribution or converge towards a dominant mode of that distribution, whereas a unimodal-output feedfoward network would converge to some statistic like the conditional expectation, which might not correspond to a coherent configuration of the output variables when the actual conditional distribution is multimodal.

This paper proposes such an approach combining a first feedforward phase with a second iterative phase corresponding to searching for a dominant mode of the conditional distribution while tackling the problem of semantic image segmentation. We take advantage of theoretical results (Alain & Bengio, 2013) on denoising autoencoder (DAE), which show that they can estimate the *score* or negative gradient of the energy function of the joint distribution of observed variables: the difference between the reconstruction and the input points in the direction of that estimated gradient. We propose to condition the autoencoder with an additional input so as to obtain the conditional score,

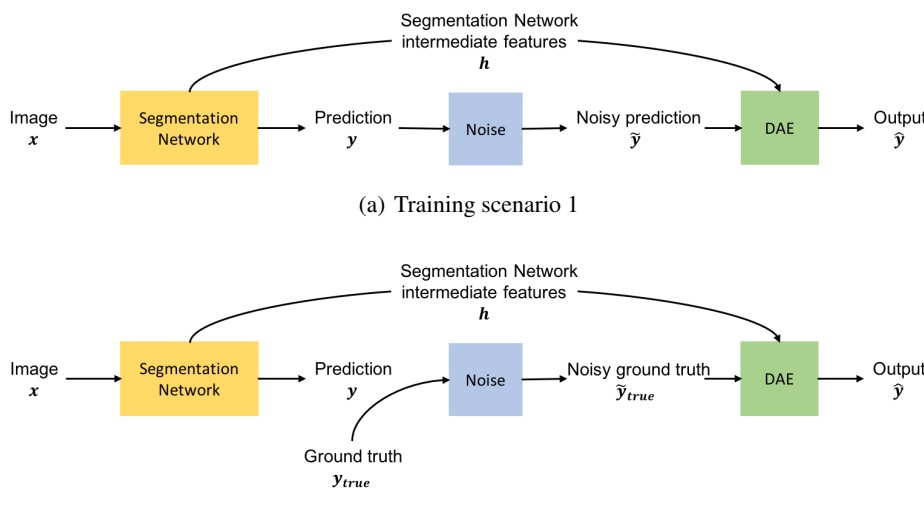

(a) Training scenario 1

(b) Training scenario 2

Figure 1: Training pipeline. Given an input image $\mathbf{x}$, we extract a segmentation candidate $\mathbf{y}$ and intermediate feature maps $\mathbf{h}$ by applying a pre-trained segmentation network. We add some noise to $\mathbf{y}$ and train a DAE that takes as input both $\mathbf{y}$ and $\mathbf{h}$ by minimizing Eq. 3. Training scenario 1 (a) yields the best results and uses the corrupted *prediction* as input to the DAE during training. Training scenario 2 (b) corresponds to the original DAE prescription in the conditional case, and uses a corruption of the ground truth as input to the DAE during training.

i.e., the gradient of the energy of the conditional density of the output variables, given the input variables. The autoencoder takes a candidate output $\mathbf{y}$ as well as an input $\mathbf{x}$ and outputs a value $\hat{\mathbf{y}}$ so that $\hat{\mathbf{y}} - \mathbf{y}$ estimates the direction $\frac{\partial \log p(\mathbf{y}|\mathbf{x})}{\partial \mathbf{y}}$. We can then take a gradient step in that direction and update $\mathbf{y}$ towards a lower-energy value and iterate in order to approach a mode of the implicit $p(\mathbf{y}|\mathbf{x})$ captured by the autoencoder. We find that instead of corrupting the segmentation target as input of the DAE, we obtain better results by training the DAE with the corrupted feedforward prediction, which is closer to what will be seen as the initial state of the iterative inference process. The use of a denoising autoencoder framework to estimate the gradient of the energy is an alternative to more traditional graphical modeling approaches, e.g., with conditional random fields (CRFs) (Lafferty et al., 2001; He et al., 2004), which have been used to model the joint distribution of pixel labels given an image (Krähenbühl & Koltun, 2011). The potential advantage of the DAE approach is that it is not necessary to decide on an explicitly parametrized energy function: such energy functions tend to only capture local interactions between neighboring pixels, whereas a convolutional DAE can potentially capture dependencies of any order and across the whole image, taking advantage of the state-of-the-art in deep convolutional architectures in order to model these dependencies via the direct estimation of the energy function gradient. Note that this is different from the use of convolutional networks for the feedforward part of the network, and regards the modeling of the conditional joint distribution of output pixel labels, given image features.

The main contributions of this paper are the following:

1. A novel training framework for modeling structured output conditional distributions, which is an alternative to CRFs, inspired by denoising autoencoder estimation of energy gradients.
2. Showing how this framework can be used in an architecture for image pixel-wise segmentation, in which the above energy gradient estimator is used to propose a highly probable segmentation through gradient descent in the output space.
3. Demonstrating that this approach to image segmentation outperforms or matches classical alternatives such as combining convolutional nets with CRFs and more recent state-of-the-art alternatives on the CamVid dataset.

## 2 METHOD

In this section, we describe the proposed iterative inference method to refine the segmentation of a feedforward network.

## 2.1 BACKGROUND

As pointed in section 1, DAE can estimate a density $p(\mathbf{y})$ via an estimator of the score or negative gradient $-\frac{\partial \mathcal{E}}{\partial \mathbf{y}}$ of the energy function $\mathcal{E}$ (Vincent et al., 2010; Vincent, 2011; Alain & Bengio, 2013). These theoretical analyses of DAE are presented for the particular case where the corruption noise added to the input is Gaussian. Results show that DAE can estimate the gradient of the energy function of a joint distribution of observed variables. The main result is the following:

$$\frac{\partial \log p(\mathbf{y})}{\partial \mathbf{y}} \approx \frac{1}{\sigma^2} \left( r(\mathbf{y}) - \mathbf{y} \right), \tag{1}$$

where $\sigma^2$ is the amount of Gaussian noise injected during training, $\mathbf{y}$ is the input of the autoencoder and $r(\mathbf{y})$ is its output (the reconstruction). The approximation becomes exact as $\sigma \to 0$ and the autoencoder is given enough capacity, training examples and training time. The direction of $(r(\mathbf{y}) - \mathbf{y})$ points towards more likely configurations of $\mathbf{y}$. Therefore, the DAE learns a vector field pointing towards the manifold where the input data lies.

## 2.2 OUR FRAMEWORK

In our case, we seek to rapidly learn a vector field pointing towards more probable configurations of $\mathbf{y}|\mathbf{x}$. We propose to extend the results summarized in subsection 2.1 and condition the autoencoder with an additional input. If we condition the autoencoder with features $\mathbf{h}$, which are a function of $\mathbf{x}$, the DAE framework with Gaussian corruption learns to estimate $\frac{\partial \log p(\mathbf{y}|\mathbf{x})}{\partial \mathbf{y}} = -\frac{\partial \mathcal{E}(\mathbf{y},\mathbf{h})}{\partial \mathbf{y}}$, where $\mathbf{y}$ is a segmentation candidate, $\mathbf{x}$ an input image and $\mathcal{E}$ is an energy function. Gradient descent in energy can thus be performed in order to iteratively reach a mode of the estimated conditional distribution:

$$\mathbf{y} \leftarrow \mathbf{y} - \epsilon \frac{\partial \mathcal{E}(\mathbf{y}, \mathbf{h})}{\partial \mathbf{y}} \tag{2}$$

with step size $\epsilon$. In addition, whereas Gaussian noise around the target $\mathbf{y}_{\text{true}}$ would be the DAE prescription for the corrupted input to be mapped to $\mathbf{y}_{\text{true}}$, this may be inefficient at visiting the configurations we really care about, i.e. those produced by our feedforward predictor, which we use to obtain a first guess for $\mathbf{y}$, as initialization of the iterative inference towards an energy minimum. Therefore, we propose that during training, instead of using a corrupted $\mathbf{y}_{\text{true}}$ as input, the DAE takes as input a corrupted segmentation candidate $\mathbf{y}$ and either the input $\mathbf{x}$ or some features $\mathbf{h}$ extracted from a feedforward segmentation network applied to $\mathbf{x}$: $\mathbf{y} = f^L(\mathbf{x}), \quad \mathbf{h} = f^l(\mathbf{x})$, where $f^k$ is a non-linear function and $l \in \{1, ..., L\}$ is the index of a layer in the feedforward segmentation network. The output of the DAE is computed as $\hat{\mathbf{y}} = r(\tilde{\mathbf{y}}, \mathbf{h})$, where $r$ is a non-linear function which is trained to denoise conditionally and $\tilde{\mathbf{y}}$ is a corrupted form of $\mathbf{y}$. During training, $\tilde{\mathbf{y}}$ is $\mathbf{y}$ plus noise, while at test time (for inference) it is simply $\mathbf{y}$ itself.

In order to train the DAE, (1) we extract both $\mathbf{y}$ and $\mathbf{h}$ from a feedforward segmentation network; (2) we corrupt $\mathbf{y}$ into $\tilde{\mathbf{y}}$; and (3) we train the DAE by minimizing the following loss

$$\mathcal{L} = ||\hat{\mathbf{y}} - \mathbf{y_{true}}||_2^2 + \mathcal{H}\left(\hat{\mathbf{y}}, \mathbf{y_{true}}\right), \tag{3}$$

where $\mathcal{H}$ is the categorical cross-entropy and $\mathbf{y_{true}}$ is the segmentation ground truth.

Figure 1(a) depicts the pipeline during training. First, a fully convolutional feedforward network for segmentation is trained. In practice, we use one of the state-of-the-art pre-trained networks. Second, given an input image $\mathbf{x}$, we extract a segmentation proposal $\mathbf{y}$ and intermediate features $\mathbf{h}$ from the segmentation network. Both $\mathbf{y}$ and $\mathbf{h}$ are fed to a DAE network (adding Gaussian noise to $\mathbf{y}$). The DAE is trained to properly reconstruct the clean segmentation (ground truth $\mathbf{y_{true}}$). Figure 1(b) presents the original DAE prescription , where the DAE is trained by taking as input $\mathbf{y_{true}}$ and $\mathbf{h}$.

Once trained, we can exploit the trained model to iteratively take gradient steps in the direction of the segmentation manifold. To do so, we first obtain a segmentation proposal $\mathbf{y}$ from the feedforward network and then we iteratively refine this proposal by applying the following rule

$$\mathbf{y} \leftarrow \mathbf{y} + \epsilon(r(\mathbf{y}, \mathbf{h}) - \mathbf{y}). \tag{4}$$

For practical reasons, we collapsed the corruption noise $\sigma^2$ into the step size $\epsilon$.

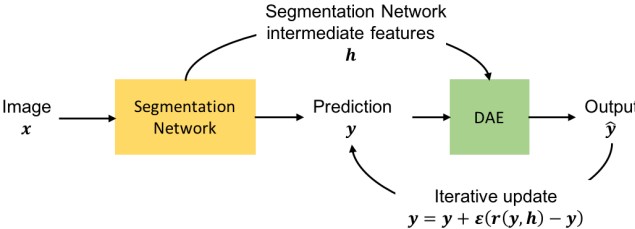

Figure 2: Test pipeline. Given an input image $\mathbf{x}$, we extract a segmentation candidate $\mathbf{y}$ and intermediate feature maps $\mathbf{h}$ by applying a pre-trained segmentation network. We then feed $\mathbf{x}$ and $\mathbf{h}$ to the trained DAE and iteratively refine $\mathbf{y}$ by applying Eq. 4. The final prediction is the last value of $\mathbf{y}$ computed in this way.

Figure 2 depicts the test pipeline. We start with an input image $\mathbf{x}$ that we feed to a pre-trained segmentation network. The segmentation networks outputs some intermediate feature maps $\mathbf{h}$ and a segmentation proposal $\mathbf{y}$. Then, both $\mathbf{y}$ and $\mathbf{h}$ are fed to the DAE to compute the output $\hat{\mathbf{y}}$. The DAE is used to take iterative gradient steps $\mathbf{y} = \mathbf{y} - \epsilon(\mathbf{y} - \hat{\mathbf{y}})$ towards the manifold of segmentation masks, with no noise added at inference time.

## 3 RELATED WORK

On one hand, recent advances in semantic segmentation mainly focus on improving the architecture design (Ronneberger et al., 2015; Badrinarayanan et al., 2015; Drozdzal et al., 2016; Jégou et al., 2017), increasing the context understanding capabilities (Gatta et al., 2014; Visin et al., 2016; Chen et al., 2015; Yu & Koltun, 2016) and building processing modules to enforce structure consistency to segmentation outputs (Krähenbühl & Koltun, 2011; Chen et al., 2015; Zheng et al., 2015). Here, we are interested in this last research direction. CRFs are among the most popular choices to impose structured information at the output of a segmentation network, being fully connected CRFs (Krähenbühl & Koltun, 2011) and CRFs as RNNs (Zheng et al., 2015) among best performing variants. More recently, an alternative to promote structure consistency by decomposing the prediction process into multiple steps and iteratively adding structure information, was introduced by (Li et al., 2016). Another iterative approach was introduced by Gidaris & Komodakis (2016) to tackle image semantic segmentation by repeatedly detecting, replacing and refining segmentation masks. Finally, the reinterpretation of residual networks (Liao & Poggio, 2016; Greff et al., 2016) was exploited by Drozdzal et al. (2017), in the context of biomedical image segmentation, by iteratively refining learned pre-normalized images to generate segmentation predictions.

On the other hand, there has recently been some research devoted to exploit results of DAE on different tasks, such as image generation (Nguyen et al., 2016), high resolution image estimation (Sønderby et al., 2017) and semantic segmentation (Xie et al., 2016). Nguyen et al. (2016) propose plug & play generative networks, which, in the best reported results, train a fully-connected DAE to reconstruct a denoised version of some feature maps extracted from an image classification network. The iterative update rule at inference time is performed in the feature space. Sønderby et al. (2017) use DAE in the context of image super-resolution to learn the gradient of the density of high resolution images and apply it to refine the output of an upsampled low resolution image. Xie et al. (2016) exploit convolutional pseudo-priors trained on the ground-truth labels in semantic segmentation task. During the training phase, the pseudo-prior is combined with the segmentation proposal from a segmentation model to produce joint distribution over data and labels. At test time, the ground truth is not accessible, thus feedforward segmentation predictions are fed iteratively to the convolutional pseudo-prior network. In this work, we exploit DAEs in the context of image segmentation and extend them in two ways, first by using them to learn a conditional score, and second by using a corrupted feedforward prediction as input during training to obtain better segmentations.

## 4 EXPERIMENTS

The main objective of these experiments is to answer the following questions: Can a conditional DAE be used successfully as the building block of iterative inference for image segmentation? Does our proposed corruption model (based on the feedforward prediction) work better than the prescribed

target output corruption? Does the resulting segmentation system outperform more classical iterative approaches to segmentation such as CRFs?

## 4.1 CamVid Dataset

CamVid[1] (Brostow et al., 2008) is a fully annotated urban scene understanding dataset. It contains videos that are fully segmented. We used the same split and image resolution as (Badrinarayanan et al., 2015). The split contains 367 images (video frames) for training, 101 for validation and 233 for test. Each frame has a resolution of 360x480 and pixels are labeled with 11 different classes.

## 4.2 Feedforward segmentation architecture

We experimented with two feedforward architectures for segmentation: the classical fully convolutional network FCN-8 of Long et al. (2015) and the more recent state-of-the-art fully convolutional densenet (FC-DenseNet103) of Jégou et al. (2017), which do not make use of any additional synthetic data to boost their performances.

**FCN-8** (Long et al., 2015): FCN-8 is a feedforward segmentation network, which consists of a convolutional downsampling path followed by a convolutional upsampling path. The downsampling path successively applies convolutional and pooling layers, and the upsampling path successively applies transposed convolutional layers. The upsampling path recovers spatial information by merging features skipped from the various resolution levels on the downsampling path.

**FC-DenseNet103** (Jégou et al., 2017): FC-DenseNet is a feedforward segmentation network, that exploits the feature reuse idea of Huang et al. (2016) and extends it to perform semantic segmentation. FC-DenseNet103 consists of a convolutional downsampling path, followed by a convolutional upsampling path. The downsampling path iteratively concatenates all feature outputs in a feedforward fashion. The upsampling path applies a transposed convolution to feature maps from the previous stage and recovers information from higher resolution features from the downsampling path of the network by using skip connections.

## 4.3 DAE architecture

Our DAE is composed of a downsampling path and an upsampling path. The downsampling path contains convolutions and pooling operations, while the upsampling path is built from unpooling with switches (also known as unpooling with index tracking) (Zhao et al., 2015; Zhang et al., 2016; Badrinarayanan et al., 2015) and convolution operations. As discussed in (Zhang et al., 2016), reverting the max pooling operations more faithfully, significantly improves the quality of the reconstructed images. Moreover, while exploring potential network architectures, we found out that using fully convolutional-like architectures with upsampling and skip connections (between downsampling and upsampling paths) decreases segmentation results when compared to unpooling with switches. This is not surprising, since we inject noise to the model's input when training the DAE. Skip connections directly propagate this added noise to the end layers; making them responsible for the data denoising process. Note that the last layers of the model might not have enough capacity to accomplish the denoising task.

In our experiments, we use DAE built from 6 interleaved pooling and convolution operations, followed by 6 interleaved unpooling and convolution operations. We start with 64 feature maps in the first convolution and duplicate the number of feature maps in consecutive convolutions in the downsampling path. Thus, the number of feature maps in the networks downsampling path is: 64, 128, 256, 512, 1024 and 2048. In the upsampling path, we progressively reduce the number of feature maps up to the number of classes. Thus, the number of feature maps in consecutive layers of the upsampling path is the following: 1024, 512, 256, 128, 64 and 11 (number of classes). We concatenate the output of 4th pooling operation in downsampling path of DAE together with the feature maps $\mathbf{h}$ corresponding to 4th pooling operation in downsampling path of the segmentation network.

## 4.4 Training and inference details

We train our DAE by means of stochastic gradient descent with RMSprop (Tieleman & Hinton, 2012), initializing the learning rate to $10^{-3}$ and applying an exponential decay of 0.99 after each

---

[1]http://mi.eng.cam.ac.uk/research/projects/VideoRec/CamVid/

Table 1: Results on CamVid dataset test set, using different segmentation networks. DAE($\mathbf{y_{true}}$) corresponds to training scenario 2 and DAE($\mathbf{y}$) corresponds to training scenario 1 from Figure 1.

| Model | Sky | Building | Pole | Road | Sidewalk | Tree | Sign | Fence | Car | Pedestrian | Cyclist | Mean IoU | Gl. accuracy |
|---|---|---|---|---|---|---|---|---|---|---|---|---|---|
| FCN-8 | 88.7 | 77.8 | 19.9 | 91.2 | 72.7 | 71.0 | 32.7 | 24.4 | 76.1 | 41.7 | 31.0 | 57.0 | 88.1 |
| FCN-8 + CRF | **90.1** | 79.2 | 17.1 | 91.7 | 74.5 | 72.2 | **36.1** | 25.1 | 77.6 | 44.3 | 32.3 | 58.2 | 89.0 |
| FCN-8 + con. mod. | **90.1** | 78.7 | 21.1 | 91.5 | 73.3 | 72.2 | 34.5 | 25.7 | 77.3 | 43.1 | 33.5 | 58.3 | 88.7 |
| FCN-8 + CRF-RNN | 88.1 | 79.4 | **22.3** | 92.0 | 75.2 | 71.8 | 31.6 | **30.1** | 76.4 | 44.7 | 34.3 | 58.7 | 88.9 |
| FCN-8 + DAE($\mathbf{y_{true}}$) | 89.0 | 78.4 | 18.5 | 91.3 | 73.3 | 71.5 | 33.3 | 24.8 | 76.7 | 43.3 | 31.6 | 57.4 | 88.4 |
| **FCN8 + DAE($\mathbf{y}$)** | 89.1 | **80.0** | 22.0 | **92.1** | 75.3 | 72.6 | 32.7 | 27.1 | **80.3** | 46.2 | **42.5** | **60.0** | **89.3** |
| FC-DenseNet | 93.1 | 83.1 | 37.8 | **94.4** | 82.3 | 77.5 | 43.9 | 37.8 | 77.2 | 59.1 | 49.5 | 66.9 | 91.5 |
| FC-DenseNet + CRF | **93.2** | **83.8** | 35.4 | 94.3 | 81.9 | **77.9** | 46.3 | **38.3** | 77.4 | 59.5 | **51.7** | 67.2 | **91.7** |
| FC-DenseNet + con. mod. | 92.6 | 83.7 | 37.9 | **94.4** | 82.2 | 77.3 | 45.0 | 37.7 | **77.4** | 60.2 | 51.0 | 67.2 | 91.6 |
| FC-DenseNet + DAE($\mathbf{y_{true}}$) | 92.7 | 83.4 | 38.0 | **94.4** | 82.2 | 77.3 | 44.6 | 37.8 | 77.3 | 59.6 | 50.5 | 67.1 | 91.5 |
| **FC-DenseNet + DAE($\mathbf{y}$)** | 93.0 | 83.6 | **38.8** | **94.4** | **82.5** | 77.7 | 44.9 | 37.8 | 77.3 | **60.3** | 50.8 | **67.4** | **91.7** |

epoch. All models are trained with data augmentation, randomly applying crops of size $224 \times 224$ and horizontal flips. We regularize our model with a weight decay of $10^{-4}$. We use a minibatch size of 10. While training, we add zero-mean Gaussian noise ($\sigma = 0.1$ or $\sigma = 0.5$) to the DAE input. We train the models for a maximum of 500 epochs and monitor the validation reconstruction error to early stop the training using a patience of 100 epochs.

At test time, we need to determine the step size $\epsilon$ and the number of iterations to get the final segmentation output. We select $\epsilon$ and the number of iterations by evaluating the pipeline on the validation set. Therefore, we try $\epsilon \in \{0.01, 0.02, 0.05, 0.08, 0.1, 0.5, 1\}$ for up to 50 iterations ($iteration \in \{1, 2, ..., 50\}$). For each iteration, we compute the mean intersection over union (mean IoU) on the validation set and keep the combination ($\epsilon$, number of iterations) that maximizes this metric to evaluate the test set.[2]

## 4.5 RESULTS

Table 1 reports our results for FCN-8 and FC-DenseNet103 without any post-processing step, applying fully connected CRF (Krähenbühl & Koltun, 2011), context network (Yu & Koltun, 2016) as trained post-processing step, CRF-RNN (Zheng et al., 2015) trained end-to-end with the segmentation network and DAE's iterative inference. For CRF, we use publicly available implementation of Krähenbühl & Koltun (2011).

As shown in the table, using DAE's iterative inference on the segmentation candidates of a feed-forward segmentation network (DAE($\mathbf{y}$)) outperforms state-of-the-art post-processing variants; improving upon FCN-8 by a margin of $3.0\%$ IoU. When applying CRF as a post-processor, the FCN-8 segmentation results improve $1.2\%$. Note that similar improvements for CRF were reported on other architectures for the same dataset (e.g. Badrinarayanan et al. (2015)). Comparable improvements are achieved when using the context module (Yu & Koltun, 2016) as post-processor ($1.3\%$) and when applying CRF-RNN ($1.7\%$). It is worth noting that our method does not decrease the performance of any class with respect to FCN-8. However, CRF loses $2.8\%$ when segmenting column poles, whereas CRF-RNN loses $1.1\%$ when segmenting signs. When it comes to more recent state-of-the-art architectures such as FC-DenseNet103, the post-processing increment on the segmentation metrics is lower, as expected. Nevertheless, the improvement is still perceivable (+ $0.5\%$ in IoU). When comparing our method to other state-of-the-art post-processors, we observe a slight improvement. End-to-end training of CRF-RNN with FC-DenseNet103 did not yield any improvement over FC-DenseNet103.

It is worth comparing the performance of the proposed approach DAE($\mathbf{y}$) with DAE($\mathbf{y_{true}}$) trained from the ground truth. As shown in the table, DAE($\mathbf{y}$) consistently outperforms DAE($\mathbf{y_{true}}$). For FCN-8, the proposed method outperforms DAE($\mathbf{y_{true}}$) by a margin of $2.2\%$. For FC-DenseNet103, differences are smaller but still noticeable. In both cases, DAE($\mathbf{y}$) not only outperforms DAE($\mathbf{y_{true}}$)

---

[2]The code to reproduce all experiments can be found here: XXXXXXXXXXXXX

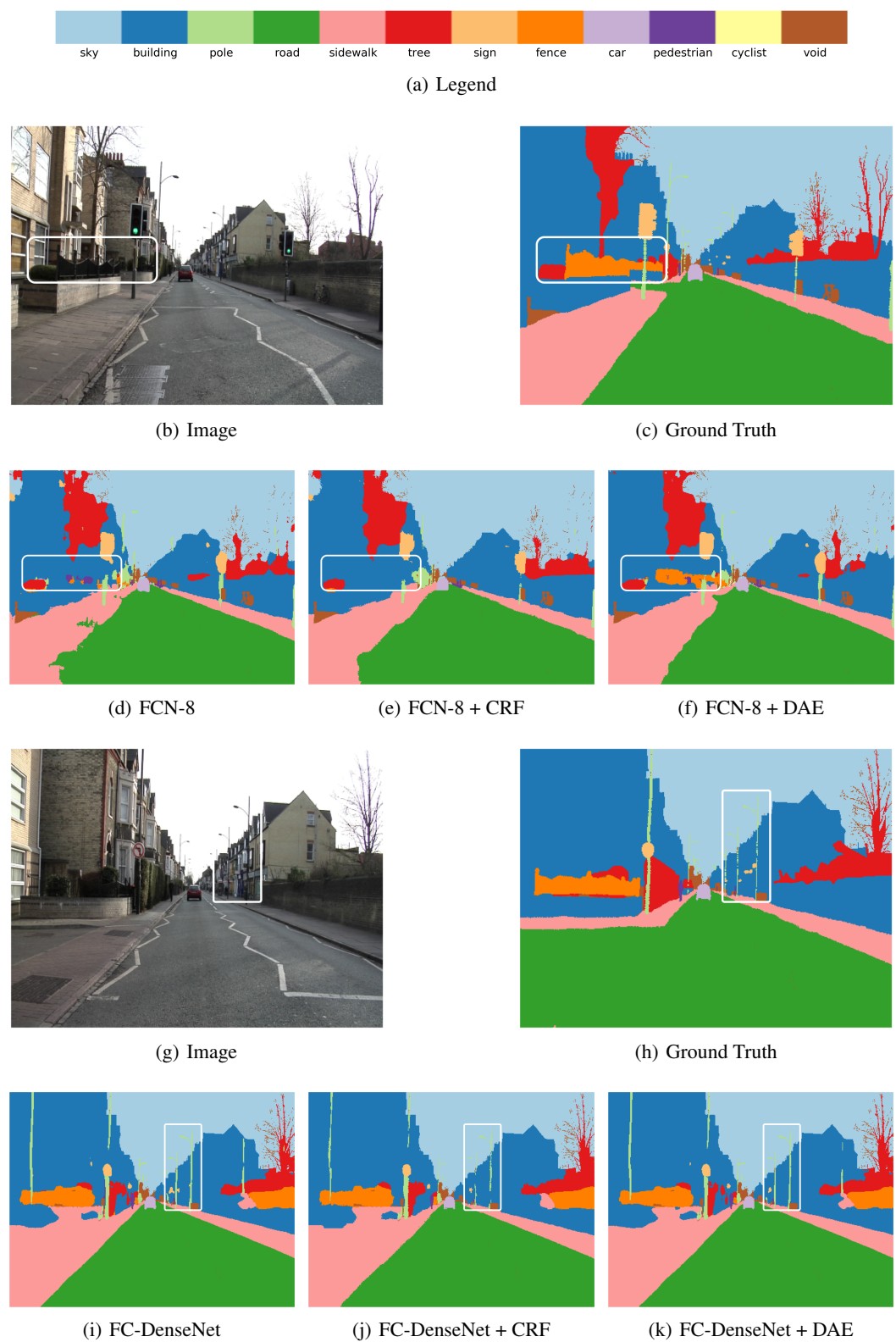

Figure 3: Qualitative results. Main differences are marked with white boxes.

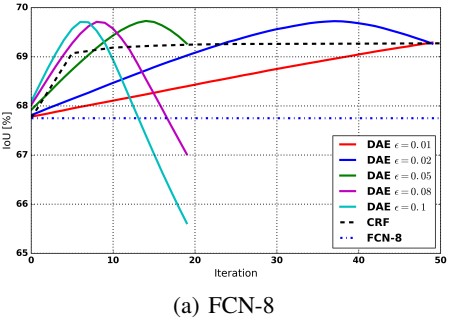

(a) FCN-8

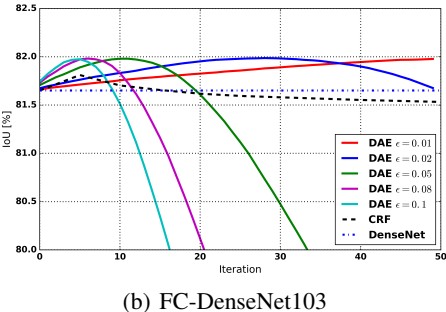

(b) FC-DenseNet103

Figure 4: Results at inference time. X-axis denotes number of iteration and Y-axis is the Intersection over Union (IoU). The plots were obtained on Camvid's validation set.

globally, but also in the vast majority of classes that exhibit an improvement. Note that the model trained on the ground truth requires a bigger Gaussian noise $\sigma$ in order to slightly increase the performance of the pre-trained feedforward segmentation networks. It is worth mentioning that training our model end-to-end with the segmentation network didn't improve the results, while being more memory demanding.

Figure 3 shows some qualitative segmentation results that compare the output of the feedforward network to both the CRF and iterative inference outputs. Figures 3(b)-3(f) show an example from the FCN-8 case, where as Figures 3(g)-3(k) show an example from FC-DenseNet103. As shown in Figure 3(d), the FCN-8 segmentation network fails to properly find the fence in the image, mistakenly classifying it as part of a building (highlighted with a white box on the image). CRF is able to clean the segmentation candidate, for example, by filling in missing parts of the sidewalk but is not able to add non-existing structure (see Figure 3(e)). Our method not only improves the segmentation candidate by smoothing large regions such as the sidewalk, but also corrects the prediction by incorporating missing objects such as the fence on Figure 3(f). As depicted in Figures 3(g)-3(k), in case of FC-DenseNet the improvement in segmentation quality is minor and difficult to perceive by visual inspection. The qualitative results follow the findings from quantitative analysis, CRF decreases slightly the quality of column pole segmentations (e. g. see area inside white boxes when comparing Figures 3(j) and 3(k)).

### 4.6 ANALYSIS OF ITERATIVE INFERENCE STEPS

In this subsection, we analyze the influence of the two inference parameters of our method, namely the step size $\epsilon$ and the number of iterations. This analysis is performed on the validation set of CamVid dataset, for the above-mentioned feedforward segmentation networks. For the sake of comparison, we perform a similar analysis on densely connected CRF; by fixing the best configuration and only changing the number of CRF iterations.

Figure 4 shows how the performance varies with number of iterations. Figure 4(a) and Figure 4(b) plot the results in the case of FCN-8 and FC-DenseNet103, respectively. As expected, there is a trade-off between the selected step size $\epsilon$ and the number of iterations. The smaller the $\epsilon$, the more iterations are required to achieve the best performance. Interestingly, all $\epsilon$ within a reasonable range lead to similar maximum performances.

## 5 CONCLUSIONS

We have proposed to use a novel form of denoising autoencoders for iterative inference in structured output tasks such as image segmentation. The autoencoder is trained to map corrupted predictions to target outputs and iterative inference interprets the difference between the output and the input as a direction of improved output configuration, given the input image.

Experiments provide positive evidence for the three questions raised at the beginning of Sec. 4: (1) a conditional DAE can be used successfully as the building block of iterative inference for image segmentation, (2) the proposed corruption model (based on the feedforward prediction) works better than the prescribed target output corruption, and (3) the resulting segmentation system outperforms state-of-the-art methods for obtaining coherent outputs.

ACKNOWLEDGMENTS

The authors would like to thank...

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
