# OpenReview forum: "Image Segmentation by Iterative Inference from Conditional Score Estimation"
_ICLR.cc/2018/Conference — Reject_

### Official Review · AnonReviewer1 · 2017-11-27

**Rating:** 5
**Confidence:** 5

**Review:**

The paper proposes an image segmentation method which iteratively refines the semantic segmentation mask obtained from a deep net. To this end the authors investigate a denoising auto-encoder (DAE). Its purpose is to provide a semantic segmentation which improves upon its input in terms of the log-likelihood.

More specifically, the authors `propose to condition the autoencoder with an additional input’ (page 1). To this end they use features obtained from the deep net. Instead of training the DAE with ground truth y, the authors found usage of the deep net prediction to yield better results.

The proposed approach is evaluated on the CamVid dataset.

Summary:
——
I think the paper discusses a very interesting topic and presents an elegant approach. A few points are missing which would provide significantly more value to a reader. Specifically, an evaluation on the classical Pascal VOC dataset, details regarding the training protocol of the baseline (which are omitted right now), an assessment regarding stability of the proposed approach (not discussed right now), and a clear focus of the paper on segmentation or conditioning. See comments below for details and other points.

Comments:
——
1. When training the DAE, a combination of squared loss and categorical cross-entropy loss is used. What’s the effect of the squared error loss and would the categorical cross-entropy on its own be sufficient? This question remains open when reading the submission.

2. The proposed approach is evaluated on the CamVid dataset which is used less compared to the standard and larger Pascal VOC dataset. I conjecture that the proposed approach wouldn’t work too well on Pascal VOC. On Pascal VOC, images are distinctly different from each other whereas subsequent frames are similar in CamVid, i.e., the road is always located at the bottom center of the image. The proposed architecture is able to take advantage of this dataset bias, but would fail to do so on Pascal VOC, which has a much more intricate bias. It would be great if the authors could check this hypothesis and report quantitative results similar to Tab. 1 and Fig. 4 for Pascal VOC.

3. The authors mention a grid-search for the stepsize and the number of iterations. What values were selected in the end on the CamVid and hopefully the Pascal VOC dataset?

4. Was the dense CRF applied out of the box, or were its parameters adjusted for good performance on the CamVid validation dataset? While parameters such as the number of iterations and epsilon are tuned for the proposed approach on the CamVid validation set, the submission doesn’t specify whether a similar procedure was performed for the CRF baseline.

5. Fig. 4 seems to indicate that the proposed approach doesn’t converge. Hence an appropriate stepsize and a reasonable number of iterations need to be chosen on a validation set. Choosing those parameters guarantees that the method performs well on average, but individual results could potentially be entirely wrong, particularly if large step sizes are chosen. I suspect this effect to be more pronounced on the Pascal VOC dataset (hence my conjecture in point 2). To further investigate this property, as a reader, I’d be curious to get to know the standard deviation/variance of the accuracy in addition to the mean IoU. Again, it would be great if the authors could check this hypothesis and report those results.

6. I find the experimental section to be slightly disconnected from the initial description. Specifically, the paper `proposes to condition the autoencoder with an additional input’ (page 1). No experiments are conducted to validate this proposal. Hence the main focus of the paper (image segmentation or DAE conditioning) remains vague. If the authors choose to focus on image segmentation, a comparison to state-of-the-art should be provided on classical datasets such as Pascal VOC, if DAE conditioning is the focus, some experiments in this direction should be included in addition to the Pascal VOC results.

Minor comment:
——
- I find it surprising that the authors choose not to cite some related work on combining deep nets with structured prediction.

---

### Official Review · AnonReviewer3 · 2017-11-27
**Good CNN based post-processing technique for semantic segmentation, but the experiments are incomplete and limited.**

**Rating:** 4
**Confidence:** 4

**Review:**

I am a returning reviewer for this paper, from a previous conference. Much of the paper remains unchanged from the time of my previous review. I have revised my review according to the updates in the paper:

Summary of the paper:
This work proposes a neural network based alternative to standard CRF post-processing techniques that are generally used on top semantic segmentation CNNs. As an alternative to CRF, this work proposes to iteratively refine the predicted segmentation with a denoising auto encoder (DAE). Results on CamVid semantic segmentation dataset showed better improvements over base CNN predictions in comparison to popular DenseCRF technique.


Paper Strengths:
- A neat technique for incorporating CRF-like pixel label relations into semantic segmentation via neural networks (auto encoders).
- Promising results on CamVid segmentation dataset with reliable improvements over baseline techniques and minor improvements when used in conjunction with recent models.


Major Weaknesses:
I have two main concerns for this work:
- One is related to the novelty as the existing work of Xie et al. ECCV'16 also proposed similar technique with very similar aim.  I think, conceptual or empirical comparisons are required to assess the importance of the proposed approach with respect to existing ones. Mere citation and short discussion is not enough. Moreover, Xie et al. seem to have demonstrated their technique on two different tasks and on three different datasets.
- Another concern is related to experiments. Authors experimented with only one dataset and with one problem. But, I would either expect some demonstration of generality (more datasets or tasks) or strong empirical performance (state-of-the-art on CamVid) to assess the empirical usefulness with respect to existing techniques. Both of these aspects are missing in experiments.


Minor Weaknesses:
- Negligible improvements with respect to CRF techniques on modern deep architectures.
- Runtime comparison is missing with respect to baseline techniques. Applying the proposed DAE 40-50 times seems very time consuming for each image.
- By back-propagating through CRF-like techniques [Zheng et al. ICCV'15, Gadde et al. ECCV'16, Chandra et al. ECCV'16 etc.], one could refine the base segmentation CNN as well. It seems this is also possible with the proposed architecture. Is that correct? Or, are there any problems with the end-to-end fine-tuning as the input distribution to DAE constantly changes? Did authors try this?


Suggestions:
- Only Gaussian noise corruption is used for training DAE. Did authors experiment with any other noise types? Probably, more structured noise would help in learning better contextual relations across pixel labels?

Clarifications:
What is the motivation to add Euclidean loss to the standard cross-entropy loss for segmentation in Eq-3?

Review summary:
The use of denoising auto encoders (DAEs) for capturing pixel label relations and then using them to iteratively refine the segmentation predictions is interesting. But, incomplete comparisons with similar existing work and limited experiments makes this a weak paper.

---

### Official Review · AnonReviewer2 · 2017-12-04
**Paper needs better experimental section and stronger baselines to validate the claimed contributions.**

**Rating:** 4
**Confidence:** 4

**Review:**

This paper proposes an iterative procedure on top of a standard image semantic segmentation networks.

The submission proposes a change to the training procedure of stacking a denoising auto-encoder for image segmentation. The technical contribution of this paper is small. The paper aims to answer a single question: When using a DAE network on top of a segmentation network output, should one condition on the predicted, or the ground truth segmentation? (why not on both?) The answer is conditioning on the predicted image for a second round of inference is a bit better. The method also performs a bit better (no statistical significance tests) than other post-processing methods (Dense-CRF, CRF-RNNs)

Experimental results are available only on a small dataset and for two different networks. This may be sufficient for a first proof-of-concept but a comparison against standard benchmark methods and datasets for semantic segmentation is missing. It is unlikely that in the current state of this submission is a contribution to image segmentation, evidence is weak and several improvements are suggested.

- The experimental evidence is insufficient. The improvements are small, statistical tests are not available. The CamVid dataset is the smallest of the image segmentation datasets used these days, more compelling would be MSCOCO or Cityscapes, better most of them. The question whether this network effect is tied to small-dataset and low-resolution is not answered. Will a similar effect be observed when compared to networks trained on way more data (e.g., CityScapes)?
- The most important baseline is missing: auto-context [Tu08]. Training the same network the DAE uses in an auto-context way. That is, take the output of the first model, then train another network using both input and prediction again for semantic segmentation (and not Eq.3). This is easy to do, practically almost always achieves better performance and I would assume the resulting network is faster and performs similar to the method presented in this submission on (guessing, I have not tried). In any case, to me this is the most obvious baseline.
- I am in favour of probabilistic methods, but the availability of an approximation of p(y) (or the nearest mode) is not used (as is most often the case).
- Runtimes are absent. This is a practical consideration which is important especially if there is little technological improvement. The DAE model of this submission compares to simple filtering methods as Krähenbühl&Koltun DenseCRF which are fast and performance results are comparable. The question wether this is practically relevant is missing, judging from the construction I guess this does not fare well. Also training time is significantly more, please comment.
- The related work is very well written, thanks. This proposal is conceptually very similar to auto-context [Tu08] and this reference missing (this is also the most important baseline)

[Tu08] Tu, “Auto-context and its application to high-level vision tasks”, CVPR 2008

---

### Decision · Program_Chairs · 2018-01-29
**ICLR 2018 Conference Acceptance Decision**

**Decision:**

Reject

**Comment:**

The experimental work was seen as one of the main weaknesses.